# Probing three-dimensional cyclooctatetraene for nucleobase modification in aptamer selection
Greta Charlotte Dahm [1], Usman Akhtar [1,2], Alix Bouvier-Müller[1], Laura Lim[1], Fabienne Levi-Acobas[1], Pierre Nicolas Bizat [1], Germain Niogret[1], Julian A. Tanner [3,4,5], Frédéric Ducongé [6] & Marcel Hollenstein [1] ✉

Decoration of aptamers with chemical modifications at the level of nucleobases grants access to alternative binding modes, which often result in improved binding properties. Most functional groups involved in such endeavours mimic the side chains of amino acids or are based on $sp^2$-dominated moieties. While this approach has met undeniable success, trends in modern drug discovery seem to favor $sp^3$-rich compounds over aromatic derivatives. Here, we report the use of a nucleotide modified with the three-dimensional, highly flexible cyclooctatetraene carboxylate (COTc). This nucleotide was engaged in an SELEX experiment against the biomarker PvLDH. Tightly binding aptamers were identified, which displayed dissociation constants in the low nM range, representing a significant improvement compared to previously identified cubamers. These modified aptamers clearly underscore the usefulness of COTc as a bioisostere replacement of aromatic moieties not only in small compounds but also in functional nucleic acids.

Aptamers are single-stranded nucleic acid sequences capable of binding to a broad variety of targets with high specificity and affinity[1–4]. These functional nucleic acids are identified by SELEX (systematic evolution of ligands by exponential enrichment) and related methods of related combinatorial methods of in vitro selection[5,6]. Given their intrinsic properties, aptamers have made a significant impact in numerous fields, including nanotechnology, sensing, and therapeutics[7–11]. The tremendous (and increasing) interest is reflected by the FDA-approval of two aptamers for the treatment of age-related macular degeneration (Pegaptanib in 2004 and avacincaptad pegol in 2023)[12]. Despite these favorable assets, aptamers consisting of natural DNA or RNA have access to a very limited number of functional groups (mainly exocyclic amines), especially when compared to protein antibodies, to mediate binding to targets. Hence, aptamers have to resort to this limited array of functional groups combined with $\pi-\pi$ stacking, hydrophobic effects (mainly via the nucleobases), hydrogen bonding, or van der Waals interactions[13]. This chemical restriction often leads to failures in SELEX or in the identification of aptamers with poor binding affinity and/or specificity. This is particularly

the case when more demanding targets (mainly of hydrophobic and anionic nature[14]) are considered, such as proteins with low (i.e., <7) pI values[15–17], highly glycosylated proteins[18–22], intrinsically disordered proteins or with little conformational definition[23–26], or small, hydrophobic molecules[27]. Chemical modification of aptamers, either during the selection protocol (mod-SELEX)[16,28–32] or after identification of binders (post-SELEX)[33–35], can remediate at least some of these shortcomings. Indeed, the addition of chemical modifications can improve binding affinities ($K_D$ values down to nM and even pM[36]), increase circulation half-lives and nuclease resistance, and convey reactivity that is not accessible to unfunctionalized DNA and RNA. Binding affinity is often increased by adding small, hydrophobic moieties, which can mimic hydrophobic contacts found in many protein-protein interactions[23,37–42]. Illustrative examples are SOMAmers (Slow Off-rate Modified Aptamers)[4,3], which are aptamers equipped with one[44–46] or multiple[47] nucleobase-modified nucleotides that display impressive binding affinities via significant reduction of $k_{off}$ rates[48]. Nonetheless, the chemical space available to SELEX is limited to a very narrow subset of functional

[1]Institut Pasteur, Université Paris Cité, CNRS UMR3523, Department of Structural Biology and Chemistry, Laboratory for Bioorganic Chemistry of Nucleic Acids, Paris Cedex 15, France. [2]Department of Pharmacy, Forman Christian College (A Chartered University), Lahore, Pakistan. [3]School of Biomedical Sciences, LKS Faculty of Medicine, The University of Hong Kong, Hong Kong, China. [4]Advanced Biomedical Instrumentation Centre, Hong Kong Science Park, Hong Kong, China. [5]Materials Innovation Institute for Life Sciences and Energy (MILES), HKU-SIRI, Shenzhen, Guangdong, China. [6]CEA, DRF, Institut of biology JACOB, Molecular Imaging Research Center (MIRCen), Université Paris Saclay, CNRS UMR9199, Fontenay aux roses, 92335, France. ✉e-mail: marcel.hollenstein@pasteur.fr

groups, mainly inspired by side chains of amino acids. Alternatively, modifications consisting of two-dimensional sp²-hybridized entities are appended to improve stacking interactions and hydrophobic contacts. Lovering et al.[49] emitted the hypothesis that including sp³ scaffolds could, amongst other benefits, increase receptor/ligand complementarity[50]. In a first step towards an escape of flatland chemistry in the aptamer World, we identified aptamers equipped with cubane-modified side chains (Fig. 1), the so-called cubamers[33,51]. These modified aptamers further validated Eaton's hypothesis that cubane **2** was a true bioisostere of benzene **1**[52] and permitted aptamers to distinguish between two closely related protein targets, i.e., the lactate dehydrogenases from *Plasmodium vivax* (PvLDH) and *Plasmodium falciparum* (PfLDH). Distinguishing PvLDH from PfLDH is of high relevance for the development of potent aptasensors since these are established malaria biomarkers[50]. Nonetheless, cubane displays steric bulk and a three-dimensional architecture but evidently lacks π character, which might be responsible for the moderate binding affinity of cubamers compared to SOMAmers ($K_D$ value of ~400 nM vs low nM range). Cyclooctatetraene (COT, **3**) is a valence isomer of cubane that displays steric bulk, π character, and a shape-shifting equilibrium that transits through a planar, antiaromatic structure of $D_{4h}$ geometry (Fig. 1)[53–58]. The non-aromatic COT has also been suggested to be capable of interacting with biomolecules in a "skeleton key" type of mechanism[55], yet few examples of such an interaction have been reported[59]. Herein, we have employed a nucleoside triphosphate equipped with a COT-carboxylate (COTc) moiety in a SELEX experiment to identify aptamers against the malaria biomarker PvLDH. We have identified three individual sequences decorated with COTc that bind to the target with very high affinity ($K_D$ values in the low nM range). This represents a substantial gain in affinity compared to corresponding unmodified and cubane-containing aptamers against the same target. The COTc-containing aptamers also highlight the capacity

of substituted cyclooctatetraene motifs to engage in interactions with biomolecules and further underscore the usefulness of three-dimensional scaffolds in aptamer selection.

## Results and Discussion

### Synthesis and biochemical characterization of the modified nucleotide

By analogy with the cubamer selection process, we designed a deoxyuridine analog equipped with a COTc moiety at position C5 of the nucleobase. As a first step in the preparation of the modified nucleotide, we converted 4-methoxycarbonylcubane-1-carboxylic acid **4**[51] into the corresponding valence isomer **5** by application of a rhodium(I) catalyzed reaction (Fig. 2)[54]. Initially, we sought to incorporate the COTc-modification by application of a CuAAC reaction with 5-ethynyl-dUTP (EdUTP) which would have allowed for a more direct comparison with cubamers. Hence, we conjugated COTc **5** with 2-azidoethylamine under standard amide bond-forming conditions and the resulting azide **7** was reacted with EdUTP to yield modified nucleotide **8** in 8% (see Supplementary Scheme 1). While nucleotide **8** acted as a good substrate for some DNA polymerases under primer extension (PEX) reaction conditions (e.g., Phusion and Q5), many reactions led to incomplete conversion of the primer to full length products and in some instances to the appearances of truncated products (Supplementary Fig. 1). In addition, enzymatic DNA synthesis under PCR conditions led to poor amplification yields especially with naïve libraries (Supplementary Fig. 2). Therefore, we had to change the design of the modified nucleotide to include a slightly longer linker arm and resorted to an amide bond as connector between COTc and the nucleobase. Such an approach also shortens the synthetic approach to a COTc-modified nucleotide. To do so, COTc **5** was directly added onto the nucleobase of commercially available amino-11-dUTP by application of standard amide bond reaction conditions (see Supplementary Figs. 8–12 for characterization of compounds)[60].

With nucleotide **6** at hand, we evaluated its compatibility with enzymatic DNA synthesis under PEX conditions and PCR. To do so, we carried out PEX reactions on a system consisting of an 18-nt long, 5'-FAM-labeled primer **P1** along with a 71-nt long template **T1** (see Supplementary Table 1 for sequence composition)[61]. We included a series of family A (Klenow fragment of *E. coli* DNA polymerase I (Kf (*exo⁻*)), Taq, Hemo KlenTaq, *Bst*), family B (Phusion, Vent (*exo⁻*), deep Vent (*exo⁻*), Therminator, Q5, phi29), and Y family (*Sulfolobus* DNA polymerase IV (Dpo4)) DNA polymerases in different PEX reactions with all dNTPs except for dTTP, which was substituted with **dU^COTcTP 6**. Gel electrophoretic analysis (PAGE 20%) of the reaction products clearly revealed that, except for phi29 all polymerases readily accepted the modified nucleotide as a substrate and produced full-length products (Fig. 3A). As expected, the modified sequences displayed a lower gel mobility due to the presence of the modifications[37,61,62]. Next, we investigated whether nucleotide **6** could also act as a substrate for polymerases under PCR conditions, which often reduces the length of the labor-intensive mod-SELEX protocol. To do so, we performed PCR with the 79-mer template **T2**[63] and primers **P2** and **P3** by using five different DNA polymerases to see whether amplicons could be produced when nucleotide **dU^COTcTP 6** substituted dTTP in the reaction mixture (Fig. 3B). This

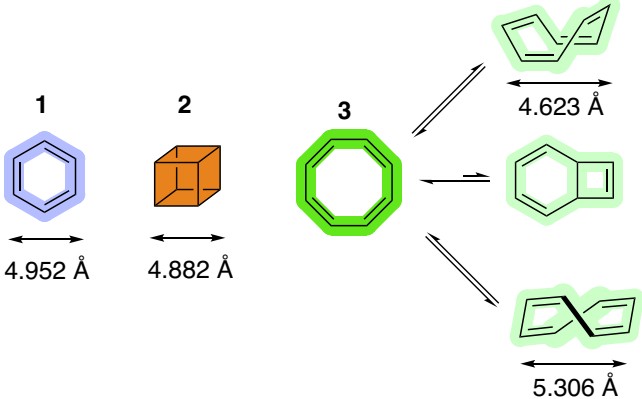

**Fig. 1 | Chemical structures of benzene (1), cubane (2), and cyclooctatetraene (3) and its valence isomers.** Bond distances are given in Å and taken from ref. 54–56. Substituted aromatic moieties have been used for the selection of SOMAmers[43], cubane **2** and substituted analogs are present in cubamers[33,51], and carboxylated COT **3** has been used in this article.

**Fig. 2 | Synthesis of dU^COTcTP 6.** Reagents and conditions: (i) [Rh(nbd)Cl]₂, Toluene, 60 °C, 20 h, 41%; (ii) a) HBTU, DIPEA, DMF, RT, 20 min; b) amino-11-dUTP, H₂O, RT, 16 h, 8%.

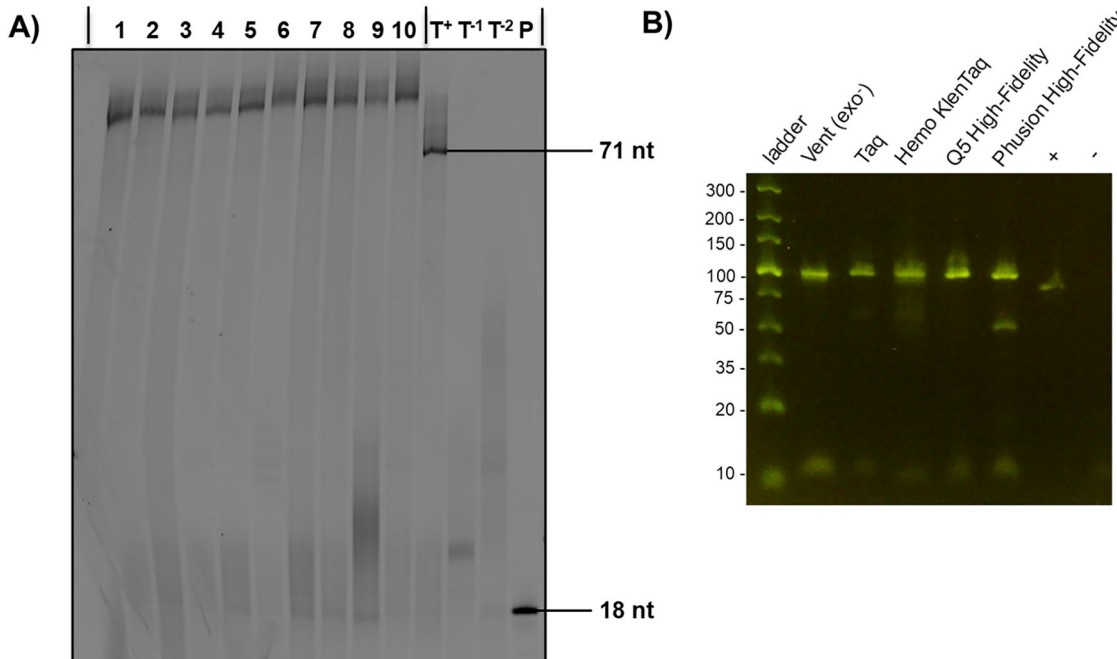

**Fig. 3 | Biochemical characterization of dU^COTc TP 6. A** Gel analysis (PAGE 20%) of primer extension reactions. The following types and quantities of polymerases were used: lane 1: Phusion (2 U), lane 2: HemoKlem Taq (8 U), lane 3: Q5 (2 U), lane 4: Bst (8 U), lane 5: Taq (5 U), lane 6: Therminator (2 U), lane 7: Vent (*exo*⁻) (2 U), lane 8: Dpo4 (2 U), lane 9: Deep Vent (2 U), lane 10: Kf (*exo*⁻) (5 U). Negative controls: Reaction mixtures containing only dATP and dGTP (T⁻¹) or dATP, dCTP, and dGTP (T⁻²) and Taq polymerase. Positive control (T⁺): with all natural nucleotides and Taq polymerase. All reactions were incubated at adequate reaction temperatures for 1 h in the presence of 200 μM of **dU^COTc TP 6. P** represents unreacted, 5'-FAM-labeled primer. **B** Agarose gel (4%) analysis of PCR products obtained with template **T2** (5'-CAC TCA CGT CAG TGA CAT GCA TGC CGA TGA CTA GTC GTC ACT AGT GCA CGT AAC GTG CTA GTC AGA AAT TTC GCA CCA C-3'), primers **P2** (5'-CAC TCA CGT CAG TGA CAT GC-3') and **P3** (5'-phosphate-GTG GTG CGA AAT TTC TGA C-3'), and a mixture of dATP, dCTP, dGTP, and **dU^COTc TP 6** (all 200 μM). Control reactions +: PCR with all four natural nucleotides and −: PCR without any polymerase and with all four natural dNTPs.

analysis revealed that nucleotide **6** was also well tolerated as a substrate by a number of polymerases under PCR conditions. Only the reaction catalyzed by Phusion produced an unidentified by-product with a faster electrophoretic mobility. Finally, we sought to evaluate whether the ester bond withstood the rather basic conditions imposed by enzymatic synthesis[64]. To do so, we carried out a PEX reaction with 20 nt long template **T3** and 5'-FAM-labeled, 19 nt long primer **P5** (see Supporting Table 1 and Supporting Protocol 1) along with **dU^COTc TP 6** and Vent (*exo*⁻). The resulting duplex equipped with a single **dU^COTc** nucleotide was then degraded by the collective action of diverse nucleases down to single nucleosides, which were then analyzed by LC-MS as described previously[61]. This LC-MS analysis clearly revealed that only the fragment corresponding to **dU^COTc** with an intact ester moiety ($m/z = 583$) was detected without detectable traces of the saponified product (Supporting Figs. 13 and 14).

**Preparation of the modified library and in vitro selection**
The biochemical analysis with nucleotide **dU^COTc TP 6** suggests that libraries for SELEX can be prepared either by PEX reactions or PCR, since various polymerases accept the modified nucleotide as a substrate under these conditions (Fig. 3). Hence, we set out to prepare a modified library suitable for SELEX by PCR. We used a library consisting of a 30 nt long randomized region flanked by two primer binding regions for PCR amplification (see Supplementary Table 1 for sequence composition). Based on the results displayed in Fig. 3B, we set out PCR experiments to evaluate the most suitable combination of reaction conditions and polymerase for amplification of the library in the presence of **dU^COTc TP 6** (Supplementary Fig. 3). PCR reactions under these conditions revealed the formation of faster running bands corresponding to side-products when Q5 was employed but not with Taq polymerase (Supplementary Fig. 3A). After optimization, we obtained good PCR amplification of the library and observed the expected shift in gel mobility compared to a naïve library obtained without modified

nucleotides (Supplementary Fig. 3B). Next, we proceeded to identify COTc-modified aptamers against PvLDH by application of a modified version of a previously reported protocol[51]. Briefly, after producing a modified dsDNA library, we removed the 5'-phosphorylated template by digestion with λ-exonuclease. The positive selection step included incubation of the resulting ssDNA naïve library with the target protein that was not conjugated to Ni-NTA-coated magnetic beads. We expected this to favor binding to the target protein[65]. After two hours of incubation, the library-protein complex was immobilized on Ni-NTA-coated magnetic particles. Eluted sequences were then reamplified using an on-beads PCR protocol[65]. The stringency of the selection protocol was controlled by including a counter-selection step against empty beads for each round of SELEX and by gradually decreasing the quantity of PvLDH (expressed and purified as described previously[33,51]). After 10 rounds of SELEX, we evaluated the binding capacity of selected, enriched pools against the protein target with an enzyme-linked oligonucleotide assay (ELONA). This analysis (Fig. 4A) clearly revealed a strong increase in binding of the libraries to PvLDH as the SELEX proceeds, thus suggesting an enrichment of the library with modified species capable of interacting with the target.

**NGS analysis of populations of the SELEX**
After ten rounds of selection, the naïve library (L0) and libraries from the 3rd, 6th and 10th rounds were sequenced by NGS to investigate if any enrichment could be observed and confirm the results obtained by ELONA. Around 100,000 reads were sequenced and analyzed per library and data were analyzed as previously described[66]. This analysis first confirmed that **dU^COTc TP 6** is an excellent substrate for polymerases since in the naïve modified library, the fraction of dT (corresponding to **6**) ranges between 20 and 30% of all bases across the 30 positions of the randomized region (see Supplementary Fig. 6). Furthermore, this analysis also shows that the frequency of dT does not drop much during the progress of the SELEX experiment and without inducing any shrinking of the length of the

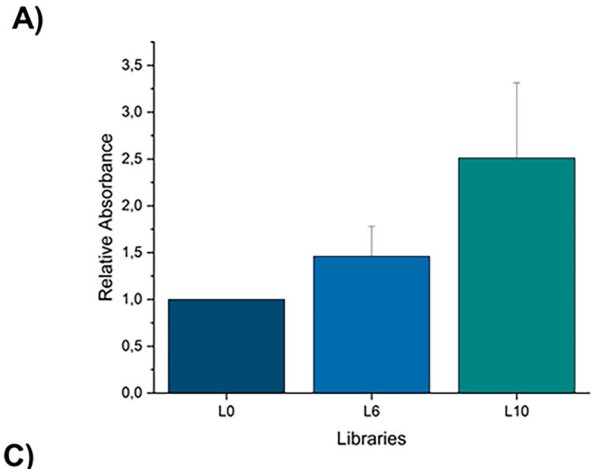

**A)**

**B)**

| Cluster | Naive library | Round 3 | Round 6 | Round 10 |
| --- | --- | --- | --- | --- |
| 0 | 0,0022 | 0,0000 | 0,0201 | 5,2695 |
| 1 | 0,0033 | 0,0000 | 0,0222 | 3,2835 |
| 2 | 0,0022 | 0,0000 | 0,0127 | 2,2667 |
| 3 | 0,0011 | 0,0000 | 0,0074 | 1,7862 |
| 4 | 0,0011 | 0,0000 | 0,0180 | 1,9706 |
| 5 | 0,0000 | 0,0000 | 0,0180 | 1,5342 |
| 6 | 0,0000 | 0,0000 | 0,0095 | 1,1536 |
| 7 | 0,0022 | 0,0000 | 0,0074 | 0,9583 |
| 8 | 0,0000 | 0,0000 | 0,0095 | 0,9862 |
| 9 | 0,0022 | 0,0000 | 0,0042 | 0,8108 |
| 10 | 0,0011 | 0,0000 | 0,0159 | 0,7928 |
| 11 | 0,0011 | 0,0011 | 0,0137 | 0,7352 |
| 12 | 0,0000 | 0,0000 | 0,0095 | 0,7028 |
| 13 | 0,0000 | 0,0000 | 0,0095 | 0,6587 |
| 14 | 0,0000 | 0,0000 | 0,0116 | 0,4949 |
| 15 | 0,0000 | 0,0000 | 0,0021 | 0,4760 |
| 16 | 0,0000 | 0,0000 | 0,0095 | 0,4553 |
| 17 | 0,0000 | 0,0000 | 0,0106 | 0,4229 |
| 18 | 0,0000 | 0,0000 | 0,0085 | 0,4211 |
| 19 | 0,0000 | 0,0000 | 0,0095 | 0,4004 |
| 20 | 0,0000 | 0,0000 | 0,0063 | 0,3590 |
| 21 | 0,0000 | 0,0000 | 0,0042 | 0,3257 |
| 22 | 0,0000 | 0,0000 | 0,0063 | 0,3095 |
| 23 | 0,0011 | 0,0000 | 0,0021 | 0,3068 |
| 24 | 0,0011 | 0,0000 | 0,0021 | 0,2960 |
| 25 | 0,0011 | 0,0011 | 0,0053 | 0,2897 |
| 26 | 0,0000 | 0,0000 | 0,0021 | 0,2664 |
| 27 | 0,0000 | 0,0000 | 0,0032 | 0,2888 |
| 28 | 0,0000 | 0,0000 | 0,0085 | 0,2439 |
| 29 | 0,0000 | 0,0000 | 0,0085 | 0,2430 |
| 30 | 0,0000 | 0,0000 | 0,0042 | 0,2349 |
| 31 | 0,0011 | 0,0000 | 0,0032 | 0,2331 |
| 32 | 0,0000 | 0,0000 | 0,0032 | 0,2268 |
| 33 | 0,0000 | 0,0011 | 0,0032 | 0,4202 |
| 34 | 0,0000 | 0,0000 | 0,0011 | 0,3176 |
| 35 | 0,0000 | 0,0000 | 0,0042 | 0,2124 |
| 36 | 0,0000 | 0,0000 | 0,0032 | 0,2124 |
| 37 | 0,0000 | 0,0011 | 0,0021 | 0,2079 |
| 38 | 0,0000 | 0,0000 | 0,0032 | 0,2052 |
| 39 | 0,0000 | 0,0000 | 0,0021 | 0,1998 |
| 40 | 0,0011 | 0,0000 | 0,0053 | 0,1989 |
| 41 | 0,0000 | 0,0000 | 0,0032 | 0,1980 |
| 42 | 0,0000 | 0,0000 | 0,0053 | 0,1971 |
| 43 | 0,0000 | 0,0000 | 0,0011 | 0,1944 |
| 44 | 0,0000 | 0,0011 | 0,0011 | 0,1800 |
| 45 | 0,0000 | 0,0011 | 0,0032 | 0,2178 |
| 46 | 0,0000 | 0,0000 | 0,0042 | 0,1755 |
| 47 | 0,0011 | 0,0011 | 0,0053 | 0,3590 |
| 48 | 0,0000 | 0,0000 | 0,0011 | 0,1728 |
| 49 | 0,0000 | 0,0000 | 0,0021 | 0,2097 |
| 50 | 0,0000 | 0,0000 | 0,0021 | 0,1620 |

**C)**

| Random sequence part | Name | Motif |
| --- | --- | --- |
| GCA**XX**ACGCAGCGC**X**GCGCAGCGAG**X**ACCG | N0 | – |
| CAG**X**GCAGACGA**X**GC**X**GGG**X**CGCCAGGGA**X** | N1 | TGGGTCGCCA |
| AA**X**CGGAA**XX**CC**X**GAGCCG**X**GCC**X**CCGAG**X** | N2 | TGAGCCG |
| CAAGCGAAG**XX**G**X**GCGCCCGCG**X**CG**X**GG**X**C | N5 | – |
| GACGGAAGAGAACGCCCACAG**X**G**X**C**X**CG**X**C | N6 | GCCCACAGTGTC |
| AGGA**XXX**CC**X**GGACC**X**GCGCGCAGG**X**CAC**X** | N7 | TTCCTGGACC |
| GCCCCG**X**CCG**X**GCAGACA**X**GCACG**XX**CCCC | N10 | GCCCCGT |
| AGG**X**CC**X**C**X**CGCCCCCAG**X**GAC**XX**GCG**X**G | N15 | GTCCTCTC |
| GCCCG**X**AC**X**CG**X**CGG**X**GGACGGCACC**X**C**X**C | N42 | GTCGGTGG |
| AA**X**GCG**X**GCCCCGC**X**GAC**X**CGA**X**GC**X**ACG**X** | N46 | GATGCTAC |

**Fig. 4 | Characterization of enrichment of the in vitro selection experiment by ELONA and HTS. A** Binding studies of three libraries by ELONA assays. Shown is the average and standard deviation of three ELONA replicates. **B** Evolution of Top50 clusters during the SELEX. This figure presents for the Top50 clusters, the frequency of the whole cluster inside each library (in %). For instance, all the sequences belonging to the cluster 0 are representing 0,0201% of the library of round 6, and 5,2695 % of the library of round 10. Clusters colored in light pink are clusters selected for the binding tests. **C** Sequences chosen for binding tests based on the NGS analysis. The exact motif present in their sequence is also given. For the sake of clarity, the primer binding regions were omitted. Red, bold-face **X** indicates the position of the modified nucleotide.

randomized region since over 90% still contain $30 \pm 1$ nucleotides in the population of round 10 (see Supplementary Fig. 6 and Supplementary Table 2). Among the 290,547 unique sequences identified, 811 sequences had a frequency that was superior to 0,01% in at least one round. These sequences were retrieved and grouped into 650 clusters according to a Levenshtein distance of 6, meaning that within one family sequences have a maximum of six mutations compared to the lead sequence. Importantly, a significant enrichment can be observed for several clusters, 82 clusters have a frequency of at least 0,1% in the library of the 10th round, while their frequency is around 100 times lower in the library of the 6th round (see Supplementary Table 3). For example, cluster 0 accounts for only 0.02% in the library of the 6th round (Fig. 4B) but for over 5% of the sequences in the 10th round library. Moreover, the alignment of the lead sequences of the top 200 clusters shows that different "motifs" are shared by several different clusters (Supplementary Table 4).

We decided to choose 10 clusters and to select their lead sequences to test their capacity to bind the PvLDH protein (Fig. 4C). These clusters were chosen according to two parameters: their enrichment inside the library and the presence of the different motifs. For each motif, we decided to select the best enriched cluster for ELONA binding assays. The majority of these enriched clusters contain between four and seven **dU**$^{COTc}$, again suggesting that dTs are not depleted during in vitro evolution.

**Binding affinity determination using flow cytometry**
Based on the NGS analysis, we performed a first screening assay of the different aptamer candidates by ELONA (see Supplementary Fig. 5). This analysis revealed that some sequences, such as N1, N2, N6, N42, and N46,

displayed little (if any) propensity at binding to PvLDH. On the other hand, other sequences such as N0, N10, or N15 might interact strongly with the target protein. However, both ELONA approaches (displayed in Fig. 4A and Supplementary Fig. 5) were impinged by non-negligible levels of errors, presumably caused by the preparation of modified sequences and libraries[67,68]. Similar outcomes of ELONA assays have previously been observed in other instances with aptamers[69]. Hence, we turned to flow cytometry to evaluate the binding capacity of the aptamer candidates. We hypothesized that flow cytometry would require only low amounts of each modified sequence, which is compatible with enzymatic production and abrogates the need for chemical synthesis. Hence, we based our analysis on a recently developed flow cytometry approach for unmodified aptamers[69]. To do so, we first produced suitable 5'-FAM-labelled, modified ssDNA sequences corresponding to the motifs displayed in Fig. 4C using PCR with primers **P1** and **P2** and 5'-phosphorylated templates (See Supplementary Fig. 4 and Supplementary Table 1). The resulting PCR products where then converted to COTc-modified ssDNA by λ-exonuclease digestion of the unmodified templates. In addition to the modified sequences N0 through N46, we prepared a control sequence (Ctrl) using a similar protocol. This control sequence (Ctrl) consists of a totally unrelated oligonucleotide of the same length as the aptamer candidates and containing 10 COTc-modifications (see Supplementary Table 1). The resulting sequences were then first incubated with PvLDH in binding buffer for 1 h. The resulting complexes were then subjected to flow cytometry analysis (Fig. 5). This analysis confirmed that most of the sequences identified by ELONA were capable of binding to target protein and that the control sequence did not bind to the target. In addition, we incubated all the modified sequences with

empty Ni-NTA beads and did not observe any significant non-specific binding activity (see Supplementary Fig. 7).

Based on this analysis, we further characterized the binding affinity of the most promising candidates, namely N0, N5, and N15. To do so, we prepared modified sequences as well as their unmodified counterparts using PCR with either a mixture of natural and modified dNTPs or only unmodified nucleotides, respectively. We then subjected the resulting sequences to a flow cytometry analysis using a range of concentrations (0, 2.5, 5, 10, 20, 60, and 100 nM). Aptamer N0 displayed the highest affinity for PvLDH with a $K_D$ value of $3.7 \pm 0.8$ nM while in the range of concentrations that were evaluated the sequence devoid of COTc-modified nucleotides did not show a typical binding curve suggesting poor and/or non-specific binding to PvLDH (Fig. 6A). A similar pattern was observed with aptamer N15 since a $K_D$ value of $5.6 \pm 1.7$ nM was determined and a non-typical binding curve was observed with the unmodified sequence with a $K_D$ value estimate of 208 nM (Fig. 6C). The shape of the binding curve of unmodified sequence N15 did not display the typical curve for cooperative, 1:1 binding and curve fitting was thus not convincing[70]. Aptamer N5 displayed a slightly reduced binding affinity compared to N0 and N15 ($K_D$ value of $14.4 \pm 7.9$ nM), but the unmodified sequence seems to be poorly to target (Fig. 6B). Similar shapes of binding curves were observed for HNA[71]. Overall, all three

aptamers, N0, N5, and N15, displayed significantly improved (>150-fold) binding affinities compared to a previously identified cubamer. All aptamers strongly depend on the presence of the modified nucleotides to interact with the target protein.

We next sought to determine whether the modified aptamers N0, N5, and N15 retained the specificity for PvLDH displayed by the cubamer. To do so, we performed a similar flow cytometry binding analysis with the modified sequences N0, N5, and N15 but with PfLDH instead of PvLDH (Fig. 7A). This analysis revealed that the sequence N15 displayed a similar propensity to interact with both proteins. A similar result was observed with N5, while N0 binds significantly less to PfLDH than PvLDH but is still capable of interacting with both proteins. Besides binding to parent protein PvLDH and PfLDH, which displays a very high sequence homology[51], we sought to assess the binding specificity against unrelated proteins. First, we carried out flow cytometry binding analysis with recombinant streptavidin (Fig. 7B). This analysis clearly revealed that none of the modified aptamers were capable of recognizing this protein as a target. Similarly, when the transpeptidase YkuD from *B. subtilis*[72–74] was used in a similar assay (Fig. 7C), modified aptamers N0 and N5 displayed no propensity to bind to this protein. The binding assay with sequence N15 displayed a slightly higher mean fluorescence intensity than that observed with the control sequence, but lower than that when PvLDH was used under the same conditions. These observations suggest a potential low, non-specific binding of N15 with the YkuD protein, most likely connected to the rather high pI value (~10) of the protein.

## Discussion

We have used in vitro selection to raise aptamers modified with cyclooctatetraene moieties that adopt a three-dimensional, saddle-like architecture. The resulting COTc-modified aptamers were raised against the malaria biomarker PvLDH to provide a direct comparison with previously identified cubamers. After ten rounds of SELEX, three COTc-modified aptamers (N0, N5, and N15) were identified that do not share any sequence homology with the cubamer but bear a similar amount (N0 and N5) or slightly more (N15) modified nucleotides. The COTc-modified aptamers displayed a superior binding affinity for the same target as the cubamer (100 to 150-fold improvement) and unmodified DNA aptamers (2-10 fold improvement)[75]. Binding experiments clearly revealed that both the presence of modifications and the exact sequence composition of N0, N5, and N15 are critically needed for activity since unmodified or unrelated sequences displayed no binding capacity. In addition, the surge in binding avidity might be a direct consequence of chemical and structural differences between cubane and COTc. Even though both structural motifs are non-planar and display similar lipophilicities[54,55], cubane only contains sp³-hybridized carbons, and COTc has a strong π-character. In addition, cubane adopts a rather rigid

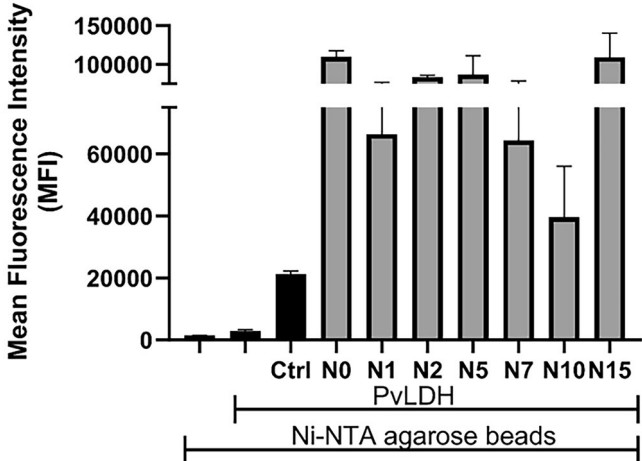

**Fig. 5 | Flow cytometry binding analysis of aptamer candidates with PvLDH.** 10 nM 5'-FAM-labeled sequences along with a control sequence (Ctrl) were incubated with His-tagged PvLDH (10 µg) in binding buffer at room temperature for 1 hour. After washing and preparation of Ni-NTA agarose beads, the beads were incubated with the aptamer-PvLDH solution for 30 minutes and analyzed by flow cytometry using the Attune NxT Flow Cytometer.

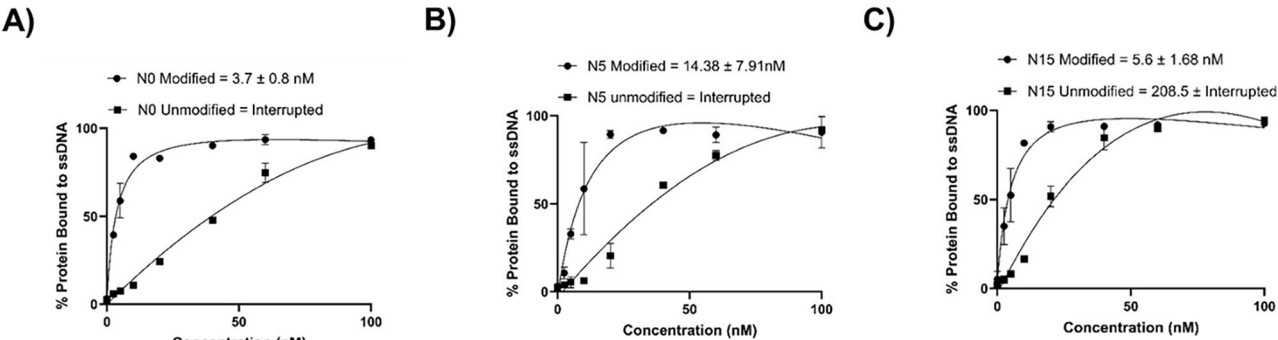

**Fig. 6 | Dissociation constant ($K_D$) determination of aptamers.** Various concentrations of 5'-FAM-labeled modified and unmodified aptamers corresponding to sequences **A** N0 ($R^2 = 0.98$), **B** N5 ($R^2 = 0.93$), and **C** N15 ($R^2 = 0.96$) were used to evaluate binding interactions. The binding curves for modified aptamers were analyzed using GraphPad Prism with nonlinear regression (curve fit) based on a one-site total binding model. The calculated $K_D$ values and their 95% confidence intervals were used for standard deviation calculations. $K_D$ values for the unmodified aptamers were found to be undefined, preventing the calculation of a mean value during the analysis.

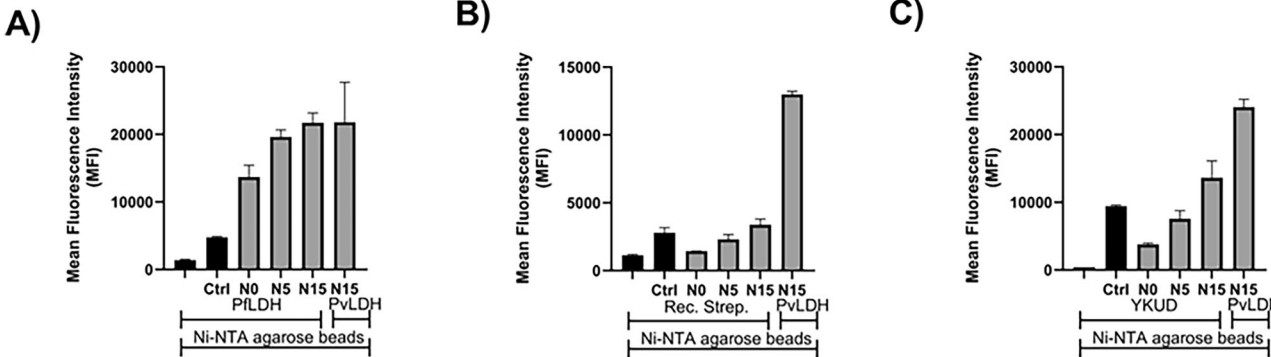

**Fig. 7 | Flow cytometry analysis of the binding specificity of aptamers against different proteins.** **A** The comparison of binding of modified sequences against PfLDH; **B** comparison of binding of modified sequences against recombinant streptavidin; **C** comparison of binding of modified sequences against transpeptidase YkuD from *B. subtilis*[72–74]. 5' FAM-labeled aptamers N0, N5, and N15 (10 nM) along with a control sequence (Ctrl) were incubated with His-tagged proteins (10 μg) in binding buffer at room temperature for 1 hour. After washing and preparation of Ni-NTA agarose beads, the beads were incubated with the aptamer-protein solutions for 30 minutes and analyzed by flow cytometry using the Attune NxT Flow Cytometer. PvLDH was used as a control for comparison of the binding interaction of aptamers.

cage structure while COTc can navigate between various structural conformations (Fig. 1). This combination of steric bulk, π-stacking capacity, and conformational flexibility makes COTc an even better phenyl ring bioisostere than cubane. In addition to these effects, direct comparison between cubamers and COTc-modified aptamers is complicated by the differential nature of the linker connecting the nucleobase to COTc/cubane as well as the presence of a carboxylate which might have an impact on the binding affinity. While no specific studies have been dedicated to this topic, it is believed that longer and less rigid linker arms reduce the efficiency of functional nucleic acids[38,76–78]. This would imply that the structural and chemical differences between COTc and cubane might be strong enough to counterbalance the negative impact of the longer and more flexible linker arm present in **dU**^COTc**TP 6**. Taken together, in addition to improving the bioactivity of small, pharmaceutical, and agrochemical compounds[54,55,59], COTc also provides aptamers with higher binding affinities. This is evidenced by the substantial improvement of dissociation constants of COTc-modified aptamers compared to cubamers and unmodified DNA aptamers.

On the other hand, COTc-modified aptamers seem to have lost the capacity of cubamers to discriminate PvLDH from PfLDH (~90% amino acid identity). This lack of specificity might arise from the absence of a negative counterselection step, including incubation with PfLDH which we have used for the identification of the cubamer. Additionally, due to ring strain, cubyl hydrogen atoms are ~$10^5$ more acidic than those present on phenyl rings and comparable to that of NH$_3$[79,80]. This acidity enables cubane derivatives to engage in non-classical C-H$_{cubane}$···O hydrogen bonding interactions[80]. Such a hydrogen bonding interaction was observed in the crystal structure of the cubamer binding to PvLDH. Importantly, this C-H$_{cubane}$···O interaction between a cubane and the carbonyl of Leu232 was believed to participate in the discrimination capacity of the cubamer. Such a hydrogen bonding capacity is obliterated by swapping cubane with COTc modifications and might be partially responsible for the lack of specificity. On the other hand, the modified COTc-aptamers did not show any propensity at binding two unrelated proteins (recombinant streptavidin and the transpeptidase YkuD), suggesting that even though cross-reactivity against PfLDH was observed, COTc-aptamers still display an important specificity.

Future work will encompass structural elucidation of the PvLDH-COTc-modified aptamer complexes. This will permit to shed light on the binding mechanism of the COTc-modified aptamers. Importantly, such work will also determine the conformational preference of the COTc moieties in this context. Indeed, it is currently unclear which geometry the COTc substituents adopt upon binding to the target. It is likely that the COTc adopts the nonplanar geometry ($D_{2d}$ symmetry) of the ground state, but other conformations (e.g., planar antiaromatic $D_{4h}$ or the bicyclic valence isomer) could be imposed upon binding to the protein[81,82]. Hence,

future structural studies of complexes of aptamers endowed with other exotic functional groups, such as COT, annulenes[83], or boroles[58,84] and protein targets such as PvLDH might potentially provide insights into specific conformers adopted by these organic moieties via trapping by binding to the target.

## Conclusions

Darwinian evolution combined with modified nucleotides represents an alluring strategy to improve the properties of functional nucleic acids. The potency of this approach is showcased by SOMAmers, which are capable of binding to targets with $K_D$ values in the low nM/high pM range[23,43,45,47] and DNAzymes capable of cleaving amide bonds[85] or hydrolyzing RNA in the absence of $M^{2+}$-cofactors[86,87]. Nonetheless, most nucleotides that have been engaged in SELEX experiments are endowed with amino acid-like residues or flat, aromatic moieties. On the other hand, modern trends in drug discovery and medicinal chemistry tend to favor $sp^3$-rich compounds[49,88]. Following this precept, we report herein the synthesis of a nucleotide modified with a cyclooctatetraene-carboxylate (COTc) moiety and its application to in vitro selection. The COTc substitution allowed to identify highly potent COTc-modified aptamers that bind to the malaria biomarker PvLDH with low nM binding affinity. This represents a significant gain in binding affinity compared to a previously reported cubamer and unmodified DNA aptamers. Even though cross-reactivity against PfLDH was observed, none of the modified aptamers bound to unrelated proteins. Taken together, these results indicate the beneficial effect of nucleotides modified with substituents capable of adopting three-dimensional conformations in aptamer selection. We also further demonstrate the usefulness of the COTc motif in bioactive molecule discovery. Future work will encompass structural elucidation of the COTc-modified aptamer-PvLDH complexes and SELEX with two nucleotides equipped with three-dimensional substituents.

## Methods
### Chemical syntheses
Detailed protocols for the synthesis of all nucleoside and nucleotide analogs can be found in the Supporting Information of this article.

### Protocol for primer extension (PEX) reactions
5'-FAM-labeled primer **P1** (10 pmol) was hybridized with the corresponding template **T1** (15 pmol) in DNase/RNase-free ultrapure water. This was achieved by elevating the temperature to 95 °C and then allowing it to gradually cool down to room temperature over an hour. Subsequently, DNA polymerase (0.5 to 1 μL), suitable reaction buffer, and the required dNTP(s) were added to yield a 10 μL reaction mixture. This mixture underwent

incubation at the polymerase-specific optimal temperature for given times. The reactions were quenched by adding 10 μL of a solution containing formamide (70%), EDTA (50 mM), bromophenol (0.1%), and xylene cyanol (0.1%). The resulting reaction mixtures were analyzed by gel electrophoresis in a denaturing 20% polyacrylamide gel, complemented with 1× TBE buffer (pH 8) and urea (7 M). Visualization of PAGE gels was performed by fluorescence imaging using a Typhoon Trio phosphorimager from Cytiva.

## Protocol for PCR

The PCR mixtures were obtained by adding primers **P2/P3** (6 μM each), template **T2** (0.1 μM), modified and natural dNTPs (200 μM), polymerase (0.4 μL), and polymerase buffer in a total volume of 20 μL. PCR cycles were dependent on the nature of the DNA polymerase: denaturation at 95 °C for 30 s, annealing at 55 °C for 30 s, and elongation at 72 °C for 60 s (Vent (*exo*⁻) DNA Polymerase, Hemo KlenTaq, and Taq DNA Polymerase) or denaturation at 98 °C for 10 s, annealing at 61 °C for 30 s, and elongation at 72 °C for 60 s (Phusion High-Fidelity DNA Polymerase and Q5 High-Fidelity DNA Polymerase). After PCR amplification with 25 cycles, the reaction products were analyzed by 4% agarose gels, supplemented with 1× E-GEL sample loading buffer (loading: 1 to 5 pmol).

## Protocol for SELEX of modified aptamers

A ssDNA library with a 30-nucleotide-long randomized region was used to prepare a naïve library for SELEX using PCR with primers **P2/P3** (6 μM each), Taq as polymerase, and in the presence of **dU^{COTᶜ}TP 6**. After removing the 5'-phosphorylated template by digestion with λ-exonuclease, we carried out a counter-selection step. To do so, the library (100 pmol) was incubated with Ni-NTA Magnetic Agarose Beads (from Jena Bioscience) for 30 min at room temperature. The unbound fraction was recovered and incubated with free *Pv*LDH for 2 h at room temperature in binding buffer (100 mM NaCl, 5 mM MgCl₂, 25 mM Tris-HCl, pH 8.0). The resulting protein-library mixture was added to fresh Ni-NTA Magnetic Agarose Beads and incubated for 30 min at room temperature. The supernatant was discarded, and the beads were washed 3 × 100 μL binding buffer before being suspended in 100 μL. We then carried out on-beads PCR amplification. To do so, the bound sequences were amplified using Taq DNA Polymerase (0.5 U/μL), a mixture of natural dATP, dCTP, and dGTP (each at 75 μM) and **dU^{COTᶜ}TP 6** (75 μM), along with Taq Standard buffer, MgSO₄ (2 mM), forward primer (**P2**, 5 μM) and 5'phosphorylated reverse primer (**P3**, 5 μM). The PCR cycles consisted of a program that started at 95 °C for 30 s, then at 55 °C for 30 s, and finally at 72 °C for 60 s. The number of cycles was adjusted after each round of selection and varied from 8 to 11 cycles. Following a purification with MinElute® PCR Purification Kit (QIAGEN), the phosphorylated strand of the dsDNA library was digested with Lambda Exonuclease (NEB) to yield an ssDNA library that can be used in a subsequent round of SELEX. The stringency of the SELEX was increased by decreasing the amount of protein over the selection rounds (for rounds 1 and 2 we used 150 μg of protein, then 100 μg of protein for rounds 3 to 7, and finally 50 μg of protein for rounds 8 to 10).

## Protocol for ELONA

For ELONA binding tests, DNA libraries of rounds 0, 6, and 10 (10 nM each) or templates corresponding to individual sequences were amplified using *Taq* DNA Polymerase (0.5 U/μL), natural dATP, dCTP, and dGTP (each at 75 μM) and **dU^{COTᶜ}TP 6** (75 μM), 5′-biotinylated forward primer P4 and 5′-phosphorylated reverse primer **P3** (0.5 μM each). The following PCR conditions were used: denaturation at 95 °C for 30 s, annealing at 55 °C for 30 s, and elongation at 72 °C for 60 s for 10 cycles. After purification with the MinElute® PCR Purification Kit (QIAGEN), the phosphorylated strands were digested with λ-exonuclease (NEB). In parallel, His₆-*Pv*LDH (2 μg/mL, 100 μL) was incubated in Ni-NTA HiSorb™ Plates (QIAGEN) wells for 1.5 h at room temperature. Wells were washed four times with PBS (0.05% Tween 20). The modified, 5'-biotinylated ssDNA libraries or individual sequences (50 nM, 50 μL) were then incubated in the protein coated wells for 1 h at room temperature. After

washing three times with binding buffer (100 mM NaCl, 5 mM MgCl₂, 25 mM Tris-HCl, pH 8.0), streptavidin-HRP (50 μL, Abcam) was added to the wells and allowed to interact for 25 min at room temperature. Wells were then washed twice with the binding buffer. Three minutes after the addition of tetramethylbenzidine (50 μL, SigmaAldrich) the reaction was stopped by adding H₂SO₄ (1 M, 50 μL). Absorption was measured at 450 nm.

## Next-generation sequencing (NGS)

For this SELEX, aliquots of the library from the naïve library and rounds 3, 6, and 10 were analyzed by NGS on an iSeq 100 Sequencing System (Illumina) as previously described[66]. Approximately 100,000 sequencing reads were analyzed for each round of SELEX with several home-made scripts that were used sequentially to analyze the results and generate the corresponding graphs (Excel and GraphPad Prism). Briefly, the primer binding sites were removed to recover only the sequences corresponding to the randomized region. Sequences having a randomized region ranging between 25 and 32 nucleotides in between the primer binding sites were recovered because it is very common for sequences to undergo deletions or insertions of a few nucleotides during SELEX. The frequency of each sequence in each round was then calculated. All sequences with a frequency of at least 0.01% in one round were retrieved and clustered into families based on a Levenstein distance of 6 (i.e., all sequences with less than six substitutions, deletions, or insertions are grouped into the same family). The frequency of each cluster was then calculated for each round (Supplementary Table S3). Multiple alignment of the Top 200 clusters was performed by MultAlin[89] and the conservation of these motifs was analyzed using MEME (Supplementary Table S4)[90].

## Protocol for flow cytometry binding analysis

To study the binding interaction of aptamers with PvLDH, 10 nM of 5'-FAM-labeled modified aptamer sequences (obtained by PCR) were mixed with 10 μg of His-tagged PvLDH in the binding buffer (25 mM Tris, pH 8.0, 100 mM NaCl, 5 mM MgCl₂) and incubated for 1 hour at room temperature. Meanwhile, Ni-NTA agarose beads were washed three times with 200 μl of binding buffer. After the final centrifugation (500 x *g*, 5 min), the beads were resuspended in the binding buffer and incubated with the aptamer-PvLDH solution for 30 minutes at room temperature to immobilize the His-tagged PvLDH. The beads were then washed three times with 200 μl of binding buffer, and after the final centrifugation, the supernatant was removed. The beads were resuspended in the buffer for FACS analysis using the Attune NxT Flow Cytometer. The data were analyzed using FlowJo software. Initial gating was carried out on empty beads (Supplementary Fig. S15).

For the determination of the dissociation constant ($K_D$) of the aptamers, various concentrations of 5'-FAM-labeled modified and unmodified aptamers (0, 2.5, 5, 10, 20, 60, and 100 nM) were used, following the same protocol for bead preparation and analysis as described above. GraphPad Prism was employed to analyze the binding curves using nonlinear regression (curve fit) based on a one-site total binding model. The $K_D$ value calculated with the 95% confidence intervals was used to determine the standard deviation (±). The $K_D$ values for the unmodified aptamers were undefined, preventing the calculation of a mean value during the analysis.

## Reporting summary

Further information on research design is available in the Nature Portfolio Reporting Summary linked to this article.

## Data availability

The authors declare that all data supporting the findings of this study are available within the article and the Supplementary Information upon reasonable request from the corresponding author.

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

## Acknowledgements
The authors gratefully acknowledge financial support from Institut Pasteur. P.N.B. and M.H. acknowledge funding from the CNRS/ANR grant PEPR MolecularXiv (ANR-22-PEXM-0002). A.B.M. and M.H. acknowledge funding via a grant from the Emergency COVID-19 Fundraising Campaign of Institut Pasteur (project AptaCoV). U.A. is thankful for funding from INSERM and Ligue Contre le Cancer (FusTarG project). G. N. gratefully acknowledges a fellowship from the doctoral school MTCI of Université Paris Cité. We thank Dr. Nolwenn Jouvenet (Institut Pasteur, Paris) for granting us access to her Flow Cytometer. We gratefully thank Dr. Frédéric Bonhomme for carrying out the LC-MS analysis.

## Author contributions
G.C.D., L.L., G.N. and P.N.B. synthesized the modified nucleotides, and G.C.D. and F.L.A. carried out the biochemical characterization of the nucleotides. G.C.D. carried out the SELEX experiment with an important contribution from F.L.A. A.B.M. and F.D. carried out the sequencing of the libraries and the bioinformatic analysis. L.L., U.A. and P.N.B. synthesized the modified and natural sequences. U.A. carried out the binding studies by flow cytometry. J.T. and M.H. designed the study, and M.H. analyzed the results and wrote the paper. All authors (G.C.D., U.A., A.B.M., L.L., F.L.A., P.N.B., G.N., J.T., F.D. and M.H.) have read and approved the final version of the manuscript.

## Competing interests
The authors declare no competing interests.
