## [Transparent Peer Review file · Communications Chemistry]

Probing three-dimensional cyclooctatetraene for nucleobase modification in aptamer selection

Corresponding Author: Dr Marcel Hollenstein

Version 0:

Reviewer comments:

Reviewer #1

(Remarks to the Author)

This is a nice work on the synthesis of cyclooctatetraene-carboxylate-linked dUTP and its use in selection of aptamers. It is undoubtedly a useful addition to the arsenal of base-modifications suitable for selections and thus the work should be published. On the other hand, there are some overstatements and some other small issues that need to be addressed before the work can be accepted.

1. the modification is not cyclooctatetraene - but it is cyclooctatetraene-carboxylate - that additional carboxylate changes a lot of things (lower hydrophobicity compared to cubane or phenyl) and that should be addressed in the discussion!
2. I personally do not like to name this type of aptamer a COTmer - compared to SOMAmers which are a whole class of aptamers with generally proven activity, this modification was (so far) used for two protein targets only...
3. The authors should perform PEX with the dUCOTTP using a short template and characterize the product by MS - this is important to show that the ester bond survives the polymerase incorporation and is not hydrolyzed to COOH [see this paper: <https://doi.org/10.1039/D2SC06718H>], that shows that the ester bonds may not always survive PEX]
4. standard negative control experiments should be added using several random sequences of the same length and same number of modifications that should not bind the target protein
5. the final aptamers' selectivity should be tested also on several unrelated proteins
6. structurally related (but reactive) trans-cyclooctene-linked dNTPs could be mentioned: <https://doi.org/10.1021/acs.bioconjchem.3c00064>

Reviewer #2

(Remarks to the Author)

This work by Hollenstein and coworkers describes the use of cyclooctatetraene (COT) modified nucleotide in SELEX to obtain the COT-modified aptamers (COTmers) against the biomarker PvLDH. The authors synthesized the triphosphate of COT-modified dUTP and confirmed its incorporation into DNA by PCR. Using the COT-modified dUTP and a random DNA library, they then performed the SELEX against the His-tagged PvLDH and analyzed the enriched clusters by NGS. The selected sequences were tested for the binding against PvLDH by two methods: enzyme-linked oligonucleotide assay (ELONA) and a flow cytometry technique. They identified several COT-modified aptamers which bind to the target with KD values in the low nM range, although their selectivity against the structurally similar PfLDH was low. The authors claim that COT modification provides aptamers with higher binding affinities as these newly acquired aptamers exhibited lower KD values compared to the unmodified aptamers and the cubane-modified aptamers (cubamers) reported by the same authors previously. The idea of adopting antiaromatic moiety into DNA aptamers is interesting, however, I have major concerns regarding the advantages of COT modification and binding affinity determination. I would recommend accepting this work only if the authors can sufficiently address the following points.

The authors claim that structural features of COT such as steric bulkiness, pi-character and D4h geometry may provide the advantages over their previously reported cubamers (Proc. Natl. Acad. Sci. U.S.A. 117, 16790-16798 (2020)). However, such structural and functional superiorities of COT modification against its isosteres (i.e., cubane and phenyl) are not supported experimentally in the present work. Throughout the manuscript, the authors emphasize the higher binding affinity of the COTmers in comparison to the previously reported cubamers and unmodified aptamers, but the sequences of each aptamer do not share any similarity. The linker structure and length on the COT-modified dUTP are considerably different from those of the previously reported cubane-modified dUTP, making their structural and functional comparison difficult. In

addition, the selection processes of each aptamer are different; one cannot rule out the possibility that lack of the counter selection with PflDH may have accounted for the higher affinity of COTmers against PvLDH while sacrificing the selectivity. Overall, there is few data that supports the structural and functional superiority of COT modification over its isosteres or other chemical moieties. The authors need to compare the binding affinity of the COTmers with the phenyl- and cubane-substituted versions of the aptamers in the same sequence context with the same linker structure. These fundamental experiments are of great importance to conclude the usefulness of the COT as a three-dimensional scaffold in aptamer selection.

The binding affinity characterization of the obtained aptamers also raises several concerns. In the initial attempt, the authors used enzyme-linked oligonucleotide assay (ELONA) to evaluate the binding ability of the aptamers. ELONA is a well-established method for the determination of the binding affinity of the aptamers as exemplified by its utility in many other studies. The authors mention that this method carried considerable errors in their study. Indeed, the standard deviation of the data in Supplementary Fig 3 seems very large. Nevertheless, the same method is utilized for assessing the binding ability of the library obtained from the different cycles of SELEX in Fig 4A, from which they conclude "This analysis (Fig. 4A) clearly revealed a strong increase in binding of the libraries to PvLDH as the SELEX proceeds (Line 167)". The authors need to discuss the cause of the errors in ELONA experiments in detail to support the validity of the presented data. It is also important to clarify what the authors mean by "despite many attempts, the error on the outcome of the ELONA assay remained important (Line 217)". Is it somehow caused by the COT modification? How about the statistical significance of the data sets?

The binding affinity determination using the flow cytometry technique also requires further investigation. In this method, the authors capture the complexes of the FAM-labelled aptamers and His-tagged PvLDH on Ni-NTA agarose beads and analyze the fluorescence intensity by flow cytometry. Metal ion interaction with oligonucleotides is a general phenomenon. A recent report describes that certain oligonucleotides bind to Ni-NTA beads (doi.org/10.1039/D3CC03349J). In fact, MFI for Ctrl (There seems no description for "Ctrl" in the manuscript. What is the sequence?) in Fig. 5 seems quite high (MFI = 20000, which is roughly 20% of N0, N5 and N15). The authors should perform control experiments with each aptamer to exclude the possibility of the non-specific binding to Ni-NTA beads.

In Fig. 6C the KD value for N15 unmodified is presented as 208.5 nM. However, according to the fitting curve in Fig 6C, the KD seems lower than this value (ca. 25 nM). The authors should describe how they calculated the KD values in detail together with the potential revision of the curve-fitting method. How did they calculate "%protein bound to ssDNA"? In addition, the binding stoichiometry data should be provided for non-biased KD determination. I suggest that authors should conduct additional binding affinity measurements using SPR or ITC for the representative COTmers along with the corresponding unmodified ones. Finally, the MFI for N15 with PvLDH in Fig.5 and Fig 7 seems considerably different (MFI = 100000 in Fig. 5 and MFI = 20000 in Fig. 7). Authors should comment on this.

Reviewer #3

(Remarks to the Author)

The authors describe their efforts in a SELEX context for the in vitro selection of aptamers modified with cyclooctatetraene moieties.

This encompasses an extensive work with a carefully carried out procedure towards discovery of a series of COT-aptamers. The work is carried out with care and well described, detailing all steps of the followed procedure. While the introduction of hydrophobic moieties into aptamers through the use of modified NTPs is not unprecedented (cfr SOMAMer development), it is, to the best of my knowledge the first example of a work towards COT containing aptamers and in this way an original study.

I consider the paper suitable for publication in Natcomms provided the following remarks are taken into account:

Remarks

• Page 2, line 75-76:

Please discuss the importance of the specifically chosen protein targets (pvLDH and PflDH) and give the rationale why exactly these are targeted with COT and cubane containing aptamers?

• Page 4, lines 123-124:

Not clear where T2 and P2/P3 can be found ? While the info is included in the SI, I would suggest to incorporate this info as well in figure 3 for reader comfort

• Considering the following statement of the authors, see page 10, lines 232-326:

"we foresee that endowing aptamers with other exotic functional groups combined with structural resolution could provide insights into long-standing questions on the conformation adopted by organic compounds such as COT, annulenes 74, or boroles"

I do not see how equipping aptamers with these exotic groups can provide information about the conformational behavior of the compounds as such. The adopted conformations will be heavily influenced by the exact conditions and binding partners, so it is highly doubtful that general conclusions on conformational aspects can be drawn from such studies.

• Concerning the following statement, see page 9, line 296-300:

"... the differential nature of the linker connecting the nucleobase to COT/cubane might have an incidence on the binding

affinity of the resulting aptamers. While no specific studies have been dedicated to this topic, it is believed that longer and less rigid linker arms reduce the efficiency of functional nucleic acids 37, 67-69. This would imply that the structural and chemical differences between COT and cubane might be strong enough to counterbalance the negative impact of the longer and more flexible linker arm present in dUCOTTP 6."

This is rather speculative. The authors should perform a comparative experiment to compare a COT modified aptamer connected through a flexible linker versus one connected through a short, less flexible linker.

- Do the authors have any idea about the binding site of the aptamer in the protein? Can any model of the aptamer structure be provided or rationalized?

Version 1:

Reviewer comments:

Reviewer #1

(Remarks to the Author)

In the revised manuscript, the authors have diligently addressed all the requests of all the reviewers, added additional experiments and improved the manuscript significantly. Now the paper is suitable for acceptance as it stands.

Reviewer #2

(Remarks to the Author)

The authors have addressed all my comments, and I believe this paper is ready for publication.

Reviewer #3

(Remarks to the Author)

The authors have satisfactorily addressed most comments of the reviewers. The extra experiments strengthen the paper and contribute to better supporting the claims.

A few items still need attention:

- * P4, new sentence: "Finally, we sought to evaluate whether the ester bond withheld the rather basic conditions imposed by enzymatic synthesis conditions"

 please revise to avoid repetition of conditions and replace whitheld by withstood

- * p10, new sentence: "In addition to these effects, direct comparison between cubamers and COTc-modified aptamers is complicated by the differential nature of the linker connecting the nucleobase to COTc/cubane as well as the presence of a carboxylate might have an incidence on the binding affinity of the resulting aptamers."

 please revise as the construction of the sentence needs attention

*

Reviewer #1:

This is a nice work on the synthesis of cyclooctatetraene-carboxylate-linked dUTP and its use in selection of aptamers. It is undoubtedly a useful addition to the arsenal of base-modifications suitable for selections and thus the work should be published. On the other hand, there are some overstatements and some other small issues that need to be addressed before the work can be accepted.

Response: We thank the reviewer for the positive assessment of our manuscript and the excellent comments to improve our work.

1. the modification is not cyclooctatetraene - but it is cyclooctatetraene-carboxylate - that additional carboxylate changes a lot of things (lower hydrophobicity compared to cubane or phenyl) and that should be addressed in the discussion!

Response: We thank this reviewer for this important comment. We have now changed the name to cyclooctatetraene-carboxylate (abbreviated as COTc) throughout the manuscript and the supporting information. We have also included the following section into the discussion as recommended:

In addition to these effects, direct comparison between cubamers and COTc-modified aptamers is complicated by the differential nature of the linker connecting the nucleobase to COTc/cubane as well as the presence of a carboxylate might have an incidence on the binding affinity of the resulting aptamers.

2. I personally do not like to name this type of aptamer a COTmer - compared to SOMAmers which are a whole class of aptamers with generally proven activity, this modification was (so far) used for two protein targets only...

Response: We thank this reviewer for this comment. We have now removed the nomenclature COTmer and replaced it with COTc-modified aptamers throughout the manuscript.

3. The authors should perform PEX with the dUCOTTP using a short template and characterize the product by MS - this is important to show that the ester bond survives the polymerase incorporation and is not hydrolyzed to COOH [see this paper: <https://doi.org/10.1039/D2SC06718H> , that shows that the ester bonds may not always survive PEX]

Response: We thank this reviewer for this important comment. We have now carried out PEX reactions with a previously reported primer/template system that permits only a single incorporation of a modified nucleotide (see e.g. RSC Chem. Biol. 2024, 5, 841). The resulting modified dsDNA was then digested down to nucleosides using the NEB digestion kit (as reported in Molecules 2022, 27, 8927 and RSC Chem. Biol. 2024, 5, 841). The resulting mixture was then subjected to LCMS analysis which revealed that the ester bond remained intact during this series of treatments. We have now included the following changes to the main manuscript:

Finally, we sought to evaluate whether the ester bond withheld the rather basic conditions imposed by enzymatic synthesis conditions.⁶⁴ To do so, we carried out a PEX reaction with 20 nt long template **T3** and 5'-FAM-labelled, 19 nt long primer **P5** (see Supporting Table 1 and Supporting Protocol 1) along with **dU^{COTc}TP 6** and Vent (*exo*). The resulting duplex equipped with a single **dU^{COTc}** nucleotide was then degraded by the collective action of diverse nucleases down to single nucleosides which were then analysed by LC-MS as described previously⁶¹. This LC-MS analysis clearly revealed that only the fragment corresponding to **dU^{COTc}** with an intact ester moiety (*m/z* = 583) was detected without detectable traces of the saponified product (Supporting Figures 13 and 14).

We have inserted a citation to the following reference (as suggested by this reviewer):

61. Niogret, G. et al. A toolbox for enzymatic modification of nucleic acids with photosensitizers for photodynamic therapy. *RSC Chem. Biol.* **5**, 841-852 (2024).

And the following changes to the supporting information:

Supporting Protocol 1

Single incorporation of dU^{COTc} via PEX reaction followed by purification and digestion of dsDNA

P5 primer (100 pmol) was annealed with template **T3** (150 pmol) in DNase/RNasefree ultrapure water. This process was carried out by first raising the temperature to 95°C, then gradually reducing it down to room temperature over a period of one hour. The resulting solution was mixed with 1 µL of Vent (*exo*) polymerase, 1 µL of Thermopol buffer and 200 µM of **dU^{COTc}TP 6** and dTTP for positive control, giving a total volume of 10 µL. Next, the combined mixture was left to incubate for a period of 30 minutes at a temperature of 60°C. The resulting products were purified using Monarch DNA Cleanup columns (5 µg), each column processing a maximum capacity of 250 pmol of product. The purified products (around 100 pmol in 10 µL) were then combined with Nucleoside Digestion Mix buffer (2 µL of 10X) and Nucleoside Digestion Mix (1 µL) in a final volume of 20 µL. The reaction mixtures were left to incubate at 37°C for one hour. Finally, the resulting products were subjected to LC-MS analysis without further purification, thus completing the process of verifying the incorporation of a **dU^{COTc}TP 6** into dsDNA.

Protocol for the detection by LC-MS of dU^{COTc} after digestion of dU^{COTc}TP 6 and dsDNA

A solution of digested dsDNA or dUCOBOH was introduced into a ThermoFisher Hypersil Gold aQ chromatography column (100 X 2.1 mm, with a particle size of 1.9 µm), maintained at a temperature of 30°C. Flow rate was set at 0.3 ml/min, and isocratic elution was performed at 1% MeCN in H₂O with 0.1% formic acid for 8 minutes, then at 100% MeCN from the 9th to the 11th minute. In positive ion mode, parent ions were fragmented using a normalized collision energy of 10% in PRM (Parallel Reaction Monitoring) mode. MS2 resolution was set at 17,500 with an AGC target of 2e5, a maximum injection time of 50 ms and an isolation window of 1.0 *m/z*. The inclusion list contained the following masses: dC (228.1), dA (252.1), dG (268.1), dT (243.1) and nucleoside **dU^{COTc}** (583.2). For detection, chromatograms of ions extracted from the base fragments (± 5 ppm) were used (112.0506 Da for dC; 136.0616 for dA; 152.0565 Da

for dG; 127.0501 Da for dT and 467.25 Da for fragmented **dU^{COTc}**. To confirm assignment (fragment ion and retention time), synthetic standards were injected beforehand.

Supplementary Figure 13. LC-MS digestion chromatogram monitored by UV detection showing the nucleoside digestion profile of dsDNA after PEX reaction with template **T3** and primer **P5**. The chromatogram reveals five distinct peaks, four of which correspond to the canonical nucleosides of the template and primer: deoxycytidine (dC), deoxyadenine (dA), deoxyguanine (dG) and deoxythymidine (dT). The additional peak represents the modified deoxyuridine **dU^{COTc} 6**.

Supplementary Figure 14. MS/MS spectrum after digestion experiment.

4. standard negative control experiments should be added using several random sequences of the same length and same number of modifications that should not bind the target protein

Response: we thank this reviewer for this comment. We apologize for the confusion that we generated concerning the controls that we have used. We would like to clarify this in our response to this reviewer and in the revised version of the manuscript. In the manuscript, we have shown that unmodified sequences corresponding to N0, N5, and N15 do not bind to the target and hence that the presence of the modifications are strictly required for activity. In addition, we have used a control sequence containing COTc-modifications corresponding to an unrelated sequence which has been identified in an unpublished and unrelated cell-SELEX experiment using natural nucleotides. This sequence is of the exact same length as sequences N0, N5, and N15 and contains slightly more modifications (10 vs 4-6). We have realized that the sequence had not been included in the supporting information in our original manuscript but fixed this omission now. Taken together, both experiments show that both COTc-modifications and the identified sequence compositions need to be present at the same time to mediate binding interactions. In order to clarify this confusion, we have taken the following actions:

- sequence composition of the control sequence has been added in the supporting information.
- the following sentences were introduced in the results section of the revised version of the manuscript:

In addition to the modified sequences N0 through N46, we prepared a control sequence using a similar protocol. This control sequence consists of a totally unrelated oligonucleotide of the same length as the aptamer candidates and containing 10 COTc-modifications (see Supplementary Table 1).

This analysis confirmed that most of the sequences identified by ELONA were capable of binding to target protein and that the control sequence did not bind to target.

- the following sentence was introduced in the discussion section of the revised manuscript:

Binding experiments clearly revealed that both the presence of modifications and the exact sequence composition of N0, N5, and N15 are critically needed for activity since unmodified or unrelated sequences displayed no binding capacity.

Finally, with all due respect, we do not believe that including additional controls would add significant information to the manuscript, especially since many studies on modified aptamers follow the same type of analysis (see e.g. ACS Chem. Biol. 2023, 18, 9, 1976; JACS 2017, 139, 40, 13977; JACS 2010, 132, 12, 4141; PNAS 2017, 114, 2898; Org. Biomol. Chem., 2017,15, 1980).

5. the final aptamers' selectivity should be tested also on several unrelated proteins

Response: We thank this reviewer for this important comment (a similar comment was also made by reviewer#2). We do agree that it is important to evaluate the specificity of the resulting aptamers especially since unlike cubamers they also bound PflDH. We have therefore carried out additional binding experiments with two totally unrelated proteins as suggested. We have used recombinant streptavidin and the transpeptidase YkuD from *Bacillus subtilis* in the FACS-based assay with all three modified aptamer sequences. As shown in the revised version of

Figure 7 (and below), none of the modified sequences displayed any real propensity at binding to either recombinant streptavidin or the transpeptidase YkuD. Some degree of non-specific binding was observed though for sequence N15 against YkuD but this is most likely connected to the rather high pI value of this protein. We have now included the following section in the results section:

Besides binding to parent protein PvLDH and PfLDH which displays a very high sequence homology,⁵¹ we sought to assess the binding specificity against unrelated proteins. First, we carried out flow cytometry binding analysis with recombinant streptavidin (Fig. 7B). This analysis clearly revealed that none of the modified aptamers were capable of recognizing this protein as a target. Similarly, when the transpeptidase YkuD from *B. subtilis*⁷²⁻⁷⁴ was used in a similar assay (Fig. 7C), modified aptamers N0 and N5 displayed no propensity at binding to this protein. The binding assay with sequence N15 displayed a slightly higher mean fluorescence intensity than that observed with the control sequence but lower than that when PvLDH was used under the same conditions. These observations suggest a potential low, non-specific binding of N15 with the YkuD protein most likely connected to the rather high pI value (~10) of the protein.

Fig. 7: Flow cytometry analysis of binding specificity of aptamers against different proteins. A) comparison of binding of modified sequences against PfLDH; B) comparison of binding of modified sequences against recombinant streptavidin; C) comparison of binding of modified sequences against transpeptidase YkuD from *B. subtilis*⁷¹⁻⁷³. 5' FAM-labeled aptamers N0, N5, and N15 (10 nM) along with a control sequence (Ctrl) were incubated with His-tagged proteins (10 µg) in binding buffer at room temperature for 1 hour. After washing and preparation of Ni-NTA agarose beads, the beads were incubated with the aptamer-protein solutions for 30 minutes and analyzed by flow cytometry using the Attune NxT Flow Cytometer. PvLDH was used as a control for comparison of binding interaction of aptamers.

And in the Discussion section:

On the other hand, the modified COTc-aptamers did not show any propensity at binding two unrelated proteins (recombinant streptavidin and the transpeptidase YkuD), suggesting that even though cross-reactivity against PfLDH was observed, COTc-aptamers still display an important specificity.

And in the conclusion section:

Even though cross-reactivity against PfLDH was observed, none of the modified aptamers bound to unrelated proteins.

We have inserted citations to the following references:

72. Magnet, S. et al. Specificity of L,D-Transpeptidases from Gram-positive Bacteria Producing Different Peptidoglycan Chemotypes*. *J. Biol. Chem.* **282**, 13151-13159 (2007).

73. Bielnicki, J. et al. B. subtilis ykuD protein at 2.0 Å resolution: insights into the structure and function of a novel, ubiquitous family of bacterial enzymes. *Proteins Struct. Funct. Genet.* **62**, 144-151 (2006).

74. Hugonneau-Beaufet, I. et al. Characterization of *Pseudomonas aeruginosa* L,d-Transpeptidases and Evaluation of Their Role in Peptidoglycan Adaptation to Biofilm Growth. *Microbiol. Spectr.* **11**, e0521722 (2023).

6. structurally related (but reactive) trans-cyclooctene-linked dNTPs could be mentioned: <https://doi.org/10.1021/acs.bioconjchem.3c00064>

Response: we have inserted a citation to this reference in the revised manuscript.

Reviewer #2:

This work by Hollenstein and coworkers describes the use of cyclooctatetrarene (COT) modified nucleotide in SELEX to obtain the COT-modified aptamers (COTmers) against the biomarker PvLDH. The authors synthesized the triphosphate of COT-modified dUTP and confirmed its incorporation into DNA by PCR. Using the COT-modified dUTP and a random DNA library, they then performed the SELEX against the His-tagged PvLDH and analyzed the enriched clusters by NGS. The selected sequences were tested for the binding against PvLDH by two methods: enzyme-linked oligonucleotide assay (ELONA) and a flow cytometry technique. They identified several COT-modified aptamers which bind to the target with KD values in the low nM range, although their selectivity against the structurally similar PflDH was low. The authors claim that COT modification provides aptamers with higher binding affinities as these newly acquired aptamers exhibited lower KD values compared to the unmodified aptamers and the cubane-modified aptamers (cubamers) reported by the same authors previously. The idea of adopting antiaromatic moiety into DNA aptamers is interesting, however, I have major concerns regarding the advantages of COT modification and binding affinity determination. I would recommend accepting this work only if the authors can sufficiently address the following points.

Response: We thank this reviewer for the careful and positive assessment of our manuscript.

The authors claim that structural features of COT such as steric bulkiness, pi-character and D_{4h} geometry may provide the advantages over their previously reported cubamers (Proc. Natl. Acad. Sci. U.S.A. **117**, 16790-16798 (2020)). However, such structural and functional superiorities of COT modification against its isosteres (i.e., cubane and phenyl) are not supported experimentally in the present work. Throughout the manuscript, the authors emphasize the higher binding affinity of the COTmers in comparison to the previously reported cubamers and unmodified aptamers, but the sequences of each aptamer do not share any similarity. The linker structure and length on the COT-modified dUTP are considerably different from those of the previously reported cubane-modified dUTP, making their structural and

functional comparison difficult. In addition, the selection processes of each aptamer are different; one cannot rule out the possibility that lack of the counter selection with PflLDH may have accounted for the higher affinity of COTmers against PvLDH while sacrificing the selectivity. Overall, there is few data that supports the structural and functional superiority of COT modification over its isosteres or other chemical moieties.

Response: We thank this reviewer for this comment and we do agree on most of the points raised. Initially, we had prepared a nucleoside triphosphate containing the exact same linker arm as that of the nucleotides used in the selection of the cubamers (i.e. via CuAAC reaction with EdUTP and COTc-azide). Surprisingly, unlike other similarly modified nucleotides (including that equipped with a cubane), this modified nucleotide was tolerated as a substrate for some polymerases under primer extension reaction conditions (e.g. Phusion, Q5), most reactions lead to incomplete conversion of the primer to product and in some instances truncated products could be observed. In addition, acceptance under PCR conditions was rather modest leading only to low amplification yields and formation of truncates. Hence, due to these considerations, we resorted to change the nature of the linker arm on the nucleotide which at the same time renders a comparison difficult. We have now included the following sections discussing these issues in the revised manuscript and the supporting information:

- in the manuscript:

Initially, we sought to incorporate the COTc-modification by application of a CuAAC reaction with 5-ethynyl-dUTP (EdUTP) which would have allowed for a more direct comparison with cubamers. Hence, we conjugated COTc **5** with 2-azidoethylamine under standard amide bond forming conditions and the resulting azide **7** was reacted with EdUTP to yield modified nucleotide **8** in 8% (see Supplementary Scheme 1). While nucleotide **8** acted as a good substrate for some DNA polymerases under primer extension (PEX) reaction conditions (e.g. Phusion and Q5), many reactions led to incomplete conversion of the primer to full length products and in some instances to the appearances of truncated products (Supplementary Fig. 1). In addition, enzymatic DNA synthesis under PCR conditions led to poor amplification yields especially with naive libraries (Supplementary Fig. 2). Therefore, we had to change the design of the modified nucleotide to include a slightly longer linker arm and resorted to an amide bond as connector between COTc and the nucleobase. Such an approach also shortens the synthetic approach to a COTc-modified nucleotide. To do so, COTc **5** was directly added onto the nucleobase of commercially available amino-11-dUTP by application of standard amide bond reaction conditions (see Supplementary Figs. 8-12 for characterization of compounds).⁶⁰

- in the supplementary information:

Supplementary Scheme 1. Synthesis of COTc-modified nucleotide **7**. Reagents and conditions: i) DIC, HOBT, DMF, 2-azidoethyl-amine, RT, 48h, 51%; ii) CuI, DIPEA, EdUTP, DMF, MeCN, RT, 3h, 8%.

Synthesis of 5-(methyl 4-((2-(1H-1,2,3-triazol-1-yl)ethyl)carbamoyl)cycloocta-1,3,5,7-tetraene-1-carboxylate)-dUTP **8**

All reactants and solvents were degassed. To a solution of CuI (4.65 mg, 0.0244 mmol, 4 eq.) in MeCN (350 μ L) and DIPEA (18.25 μ L, 0.11 mmol, 36 eq.), methyl-4-((2-azidoethyl)carbamoyl)cycloocta-1,3,5,7-tetraene-1-carboxylate **7** (3.15 mg, 0.0122 mmol, 2 eq.) dissolved in DMF (150 μ L) was added. 5-ethynyl-dUTP (3.00 mg, 0.0061 mmol, 1 eq.) dissolved in H₂O (35 μ L) was added. The resulting mixture was incubated at 25°C and shaken at 1500 rpm for 3 h. The solvent was removed under reduced pressure and the product was redissolved in water. The crude product was purified via anion exchange HPLC using tetraethylammonium bromide (TEAB, 10 mM) in a gradient from 0% to 100% TEAB (1M). The desired product was obtained as a white powder (0.40 mg, 0.52 μ mol, 8%).

HR-MS (ESI-): m/z calculated for C₂₄H₂₈N₆O₁₇P₃⁻ = 765.0729 [M-H]⁻, found 765.0729.

Supplementary Figure 1. Gel analysis (PAGE 20%) of primer extension reactions. The following types and quantities of polymerases were used: lane 1: Phusion (2 U), lane 2: HemoKlem Taq (8 U), lane 3: Q5 (2 U), lane 4: Bst (8 U), lane 5: Taq (5 U), lane 6: Terminator (2 U), lane 7: Vent (*exo*⁻) (2 U), lane 8: Dpo4 (2 U), lane 9: Deep Vent (2 U), lane 10: Kf (*exo*⁻) (5 U). Negative controls: Reaction mixtures containing no polymerase (T₁), only dATP and dGTP (T₂), or dATP, dCTP, and dGTP (T₃) and Taq polymerase. Positive control (T⁺): with all natural nucleotides and Taq polymerase. All reactions were incubated at adequate reaction temperatures for 1 h in the presence of 200 μM of modified nucleotide **8**.

Supplementary Figure 2. A) Agarose gel (4%) analysis of PCR products obtained with a naïve library as a template, primers **P2** and **P3**, and a mixture of natural nucleotides only (i.e. dATP, dCTP, dGTP, and dTTP (all 200 μ M)) with either 8 or 10 PCR cycles and Taq polymerase. B) Agarose gel (4%) analysis of PCR products obtained with a naïve library as a template, primers **P2** and **P3**, and Taq or Vent (*exo*⁻) polymerases and using a mixture of unmodified dNTPs (i.e. dATP, dCTP, and dGTP) and modified nucleotide **8** (all at 200 μ M).

In addition to these modifications, we do agree that the direct comparison is difficult due to differences in linker arm and the presence of the carboxylate on the COTc (also see response to reviewer#1). The main goal of our article was merely to demonstrate that COT could be used as a three-dimensional modification in SELEX experiments. We have consequently down-toned the comparison with cubane-modified sequences throughout the manuscript.

The authors need to compare the binding affinity of the COTmers with the phenyl- and cubane-substituted versions of the aptamers in the same sequence context with the same linker structure. These fundamental experiments are of great importance to conclude the usefulness of the COT as a three-dimensional scaffold in aptamer selection.

Response: We thank this reviewer for this interesting comment. We do agree that making such analogs would be crucial to maintain a direct comparison between cubamers and the corresponding isosteres phenyl- and COT. As stated in our response to the previous comment by this reviewer, such a direct comparison was not possible due to the rather surprisingly low substrate acceptance by the corresponding modified nucleotide with a triazole connecting moiety. Hence, direct comparison would require synthesis of the corresponding phenyl- and cubyl-modified nucleotides obtained by amide bond formation on a longer linker arm followed by re-selection against PvLDH and comparison of the best binders. Undoubtedly, this would be highly valuable information but this tremendous amount of work is beyond the scope of the present work. Nonetheless, we thank this reviewer for the very useful suggestion, and we will carry out such selection experiments in the near future. Alternatively, phenyl- and cubyl-modified nucleotides that would be prepared with a longer linker arm could be incorporated in the sequences N0, N5, and N15 identified with the COTc-modifications. The resulting modified sequences could then be assayed for their capacity at binding to PvLDH. While this represents certainly less work than reselection of two series of modified aptamers, the outcome is far from certain. Indeed, the binding mechanism of base-modified often involves compact three-dimensional folds where the modifications are positioned at precise locations to interact either with the target and/or with part of the nucleotidic scaffold of neighboring (or even distal) nucleotides (see e.g. J. Biol. Chem. 2014, 289, 8720; J. Mol. Biol. 2021, 433, 167227; PNAS 2020, 117, 16790-16798; Nat. Commun. 2017, 8, 810). Hence, even small perturbations of the chemical nature of the modifications can have a highly detrimental effect on the binding efficiency and/or specificity. This is also why post-SELEX campaigns can be quite work intensive and uncertain. We thus believe that interchanging the COTc with phenyl and cubyl in N0, N5, and N15 will have a deleterious effect on binding affinity for reasons that are not necessarily connected to the modification but rather to the change in the nature of chemical modification in terms of size, shape, and polarity. We have recently reported similar findings on substituted cubanes or related bioisosteres when incorporated into the scaffold of the cubamer (ChemBioChem 2024, 25, e202300539).

The binding affinity characterization of the obtained aptamers also raises several concerns. In the initial attempt, the authors used enzyme-linked oligonucleotide assay (ELONA) to evaluate the binding ability of the aptamers. ELONA is a well-established method for the determination of the binding affinity of the aptamers as exemplified by its utility in many other studies. The authors mention that this method carried considerable errors in their study. Indeed, the standard deviation of the data in Supplementary Fig 3 seems very large. Nevertheless, the same method is utilized for assessing the binding ability of the library obtained from the different cycles of SELEX in Fig 4A, from which they conclude “This analysis (Fig. 4A) clearly revealed a strong increase in binding of the libraries to PvLDH as the SELEX proceeds (Line 167)”. The authors need to discuss the cause of the errors in ELONA experiments in detail to support the validity of the presented data. It is also important to clarify what the authors mean by “despite many attempts, the error on the outcome of the ELONA assay remained important (Line 217)”. Is it somehow caused by the COT modification? How about the statistical significance of the data sets?

Response: We thank this reviewer for this comment. Indeed, we totally agree with this reviewer that ELONA is an established method that can be used to determine K_d values of aptamers. We have carried out an ELONA assay to evaluate whether we observe an increase of fluorescence over the rounds of SELEX which is exactly what Figure 4A shows. We did not intend to use this assay for any other, quantitative or even qualitative purpose. Sequence enrichment and binding capacity hinted by Figure 4A is further confirmed by NGS sequencing data. We were also puzzled by the rather large standard deviation stemming from three independent ELONA assays. We believe that these arise due to the following reasons: 1) ELONA assays displayed in Figure 4A were carried out on (enriched) libraries and not on single, individual sequences as typically used for K_d value determinations. Due to the complexities of the libraries and the limited number of individual molecules, we believe that the errors might be quite large on libraries (see e.g. *Appl Microbiol. Biotechnol.* 2015, 99, 9791-9803; *Anal. Methods*, 2020, 12, 3823-3835); 2) heterogeneity of the library often requires fine-tuning of the conditions and might cause substantial variation from one experiment to the other. This is particularly true since the efficiency of the conversion of dsDNA to ssDNA via λ -exonuclease digestion might be affected by the number and position of modified side-chains in the duplex; and 3) the purification steps included in the preparation of the ssDNAs might reduce the amount of material present during ELONA due to adsorption of rather hydrophobic COTc-modified sequences on the spin columns. In addition, the choice of an adequate method to monitor progress of a SELEX experiment is not obvious (*Nat. Rev. Methods Primers* 2023, 3, 54), especially with modified sequences. Since Figure 4A suggested that binding improved from round 0 to round 10, we did not evaluate any other methods and sent the libraries for NGS analysis.

Concerning Supplementary Figure 5, we believed that an ELONA assay with all individual aptamer candidates would represent a fast and cheap method to identify the best binders. Also here, errors are probably connected to the preparation of modified ssDNA. Similar outcomes have recently been observed on unmodified aptamers (*Int. J. Mol. Sci.* 2024, 25, 4642).

Since the results of this ELONA investigation were not fully conclusive, we decided to opt for the development of a FACS-based approach. This approach turned out to be more robust, but also more time-consuming and required more material. Nonetheless, none of the ELONA assays did alter the conclusions of this manuscript. We merely included this in the discussion to show our different attempts but also the limits of both methods. We have changed the sentence mentioned by this reviewer to the following:

However, both ELONA approaches (displayed in Fig. 4A and Supplementary Fig. 5) were impinged by non-negligible levels of errors, presumably caused by the preparation of modified sequences and libraries^{67, 68}. Similar outcomes of ELONA assays have previously been observed in other instances with aptamers⁶⁹.

We have also inserted citations to the following references:

67. DeRosa, M.C. et al. In vitro selection of aptamers and their applications. *Nat. Rev. Methods Primers* **3**, 54 (2023).

68. Mohammadinezhad, R., Jalali, S.A.H. & Farahmand, H. Evaluation of different direct and indirect SELEX monitoring methods and implementation of melt-curve analysis for rapid discrimination of variant aptamer sequences. *Anal. Methods* **12**, 3823-3835 (2020).

69. Civit, L. et al. A Multi-Faceted Binding Assessment of Aptamers Targeting the SARS-CoV-2 Spike Protein. *Int. J. Mol. Sci.* **25**, 4642 (2024).

The binding affinity determination using the flow cytometry technique also requires further investigation. In this method, the authors capture the complexes of the FAM-labelled aptamers and His-tagged PvLDH on Ni-NTA agarose beads and analyze the fluorescence intensity by flow cytometry. Metal ion interaction with oligonucleotides is a general phenomenon. A recent report describes that certain oligonucleotides bind to Ni-NTA beads (doi.org/10.1039/D3CC03349J). In fact, MFI for Ctrl (There seems no description for "Ctrl" in the manuscript. What is the sequence?) in Fig. 5 seems quite high (MFI = 20000, which is roughly 20% of N0, N5 and N15). The authors should perform control experiments with each aptamer to exclude the possibility of the non-specific binding to Ni-NTA beads.

Response: We thank this reviewer for this comment. First of all, we would like to mention that the referenced paper (doi.org/10.1039/D3CC03349J) reported the first aptamer that recognizes Ni-NTA and determined its K_d value. The authors aimed to identify an aptamer for PD-L1 and eventually found one that binds to beads instead. However, this is a common outcome when stringent selection pressure is omitted during the selection procedure, leading to non-specific binders. In such cases, the selection process is considered unsuccessful, or the identified aptamer from NGS binds non-specifically, meaning it does not meet the definition of an aptamer. In our study, we designed a SELEX procedure with stringent negative selection steps against empty beads to eliminate sequences that would bind to empty beads only. We included a control sequence (Ctrl) with a randomized sequence of the same nucleotide length, containing same or approximately $\pm 2-3$ modifications per aptamer sequence, to compare binding in the presence of beads and protein (also see our response to comment#4 by reviewer#1). Nonetheless, we carried out additional experiments to demonstrate that the modified sequences were not binding to empty beads. Our control data showed no binding to beads alone for any of the modified sequences (with approximately 5500 MFI of N15 with beads only, which is non-significant when compared to N15 with beads and protein, see main figures of the manuscript). The roughly 20% MFI observed in the Ctrl sequence may be due to interactions between the modifications and the protein, as the modified aptamer itself did not interact with the beads. Therefore, the MFI intensity observed for the Ctrl sequence is not significant when compared to all modified control aptamer sequences.

Supplementary Figure 7. Flow cytometry binding analysis of aptamer candidates with Ni-NTA agarose beads. 10 nM 5'- FAM-labelled sequences along with a control sequence (Ctrl) were incubated with Ni-NTA agarose beads for 30 minutes and analyzed by flow cytometry using the Attune NxT Flow Cytometer.

In Fig. 6C the K_D value for N15 unmodified is presented as 208.5 nM. However, according to the fitting curve in Fig 6C, the K_D seems lower than this value (ca. 25 nM). The authors should describe how they calculated the K_D values in detail together with the potential revision of the curve-fitting method. How did they calculate “%protein bound to ssDNA”? In addition, the binding stoichiometry data should be provided for non-biased K_D determination. I suggest that authors should conduct additional binding affinity measurements using SPR or ITC for the representative COTmers along with the corresponding unmodified ones. Finally, the MFI for N15 with PvLDH in Fig.5 and Fig 7 seems considerably different (MFI = 100000 in Fig. 5 and MFI = 20000 in Fig. 7). Authors should comment on this.

Response: We thank the reviewer for this important comment. As stated in the manuscript, the binding curves for modified aptamers were analyzed using GraphPad Prism with nonlinear regression (curve fitting) based on a one-site total binding model. The calculated K_d values and their 95% confidence intervals were used to determine standard deviations. However, values for unmodified aptamers were undefined, preventing the calculation of a mean value during analysis. As described above, we used a software-based method for K_d determination, applying nonlinear regression with the values listed in the table calculated by the software for our study. We have realized that the R^2 values for the curve fitting were missing in the caption of Figure 7 in the original manuscript and have now provided these values.

The percentage of protein bound to ssDNA was calculated using FlowJo software for flow cytometry data analysis. Briefly, in FlowJo, we gated the beads protein complex as a reference. Any events above this gate were considered bound complexes, indicating successful ssDNA-protein binding.

The MFI intensity of N15 differs between Figures 5 and 7 because we decreased the laser power during the second measurement. This adjustment was necessary for K_d determination, as the 100 nM concentration produced an excessively strong signal that exceeded the graph's limits. Lowering the laser intensity ensured that all concentration values fell within the measurable range in both the MFI and laser graphs in the flow cytometer. Following this adjustment, we used the same lower laser power for all further analyses to maintain consistency across measurements.

Concerning the unmodified sequence of N15, we do agree that this sequence displays some degree of unspecific binding. However, K_d determination is delicate in this instance since no

reliable curve fitting could be obtained with the different models that were evaluated (as indicated by “interrupted” in the caption of Figure 6). This can certainly be ascribed to the shape of the curve which markedly deviates from typical, 1:1 cooperative binding curves (see Analytical Biochemistry 2020, 600, 113742). We have improved the description of the binding curves in the revised version of our manuscript and inserted a citation to this reference (#69 of the revised manuscript). The advantage of our FACS-based method, which had never been employed for modified aptamers, is that little material is necessary, compatible with enzymatic synthesis. Other biophysical methods suggested by this reviewer require preparation of the corresponding phosphoramidite followed by solid-phase synthesis of the sequences. The goal of our study was to highlight that other three-dimensional, anti/non-aromatic moieties can be employed in SELEX to raise potent binders against a protein target. More in depth biophysical and structural investigations of these COTc-modified sequences will be reported in a follow-up article.

Reviewer #3:

The authors describe their efforts in a SELEX context for the in vitro selection of aptamers modified with cyclooctatetraene moieties. This encompasses an extensive work with a carefully carried out procedure towards discovery of a series of COT-aptamers. The work is carried out with care and well described, detailing all steps of the followed procedure. While the introduction of hydrophobic moieties into aptamers through the use of modified NTPs is not unprecedented (cfr SOMAMer development), it is, to the best of my knowledge the first example of a work towards COT containing aptamers and in this way an original study. I consider the paper suitable for publication in Natcomms provided the following remarks are taken into account:

Response: We thank this reviewer for the positive and careful assessment of our manuscript.

• Page 2, line 75-76: Please discuss the importance of the specifically chosen protein targets (pVLDH and PflDH) and give the rationale why exactly these are targeted with COT and cubane containing aptamers?

Response: We thank this reviewer for this comment. We have mentioned that P_vLDH is a malaria biomarker but we have added the following sentence in the revised manuscript to clarify this point:

Distinguishing P_vLDH from PflDH is of high relevance for the development of potent aptasensors since these are established malaria biomarkers.⁵⁰

• Page 4, lines 123-124: Not clear where T2 and P2/P3 can be found ? While the info is included in the SI, I would suggest to incorporate this info as well in figure 3 for reader comfort

Response: Thank you for this comment. We have now included the sequence information of the template and both primers in the caption of Figure 3 as recommended.

• Considering the following statement of the authors, see page 10, lines 232-326:

“we foresee that endowing aptamers with other exotic functional groups combined with structural resolution could provide insights into long-standing questions on the conformation adopted by organic compounds such as COT, annulenes 74, or boroles”

I do not see how equipping aptamers with these exotic groups can provide information about the conformational behavior of the compounds as such. The adopted conformations will be heavily influenced by the exact conditions and binding partners, so it is highly doubtful that general conclusions on conformational aspects can be drawn from such studies.

Response: We thank this reviewer for this comment. We do agree that this statement is confusing. We meant to say that potentially one or multiple conformers of such organic compounds could be trapped via binding to the protein target and that structural resolution could provide insights into these particular conformations. This is obviously, as mentioned by this reviewer, dependent on the binding partner and is thus not general. We have modified the sentence to the following in order to remove this ambiguous statement:

Hence, future structural studies of complexes of aptamers endowed with other exotic functional groups such as COT, annulenes⁸³, or boroles^{58, 84} and protein targets such as PvLDH might potentially provide insights into specific conformers adopted by these organic moieties via trapping by binding to the target.

• Concerning the following statement, see page 9, line 296-300:

“... the differential nature of the linker connecting the nucleobase to COT/cubane might have an incidence on the binding affinity of the resulting aptamers. While no specific studies have been dedicated to this topic, it is believed that longer and less rigid linker arms reduce the efficiency of functional nucleic acids 37, 67-69. This would imply that the structural and chemical differences between COT and cubane might be strong enough to counterbalance the negative impact of the longer and more flexible linker arm present in dUCOTTP 6.”

This is rather speculative. The authors should perform a comparative experiment to compare a COT modified aptamer connected through a flexible linker versus one connected through a short, less flexible linker.

Response: We thank this reviewer for this important comment. Reviewer #2 made a similar comment, and we refer to our response to that comment. Briefly, we had initially prepared a nucleotide equipped with a more rigid triazole linker arm in analogy with the nucleotide used in the cubane-SELEX experiment. Surprisingly, this nucleotide was not very well-tolerated by polymerases (or at least not sufficiently to be used in SELEX) under PEX reaction conditions and PCR, which prompted us to change the design of the linker arm.

• Do the authors have any idea about the binding site of the aptamer in the protein? Can any model of the aptamer structure be provided or rationalized?

Response: We thank this reviewer for this very interesting question. For the time being, we do not have any clear proof on the structure of the aptamer nor of the binding site. We can only guess that the aptamer binds in the same hydrophobic pocket than the cubamer. We are currently investigating the possibility of using cryo-EM to resolve the structure of the COTc-modified aptamers but also the cubamer and an unmodified DNA aptamer in complex with PvLDH. As mentioned in the conclusion section of the manuscript, such structural elucidations will also be of importance to visualize the conformation of the cyclooctatetraene during binding (i.e. D_{2d}, D_{4h}, or even D_{8h}). Structural studies will be reported in due course.